# Histone demethylase AMX-1 is necessary for proper sensitivity to interstrand crosslink DNA damage

Xiaojuan Zhang[1], Sisi Tian[1], Sara E. Beese-Sims[2], Jingjie Chen[1], Nara Shin[2], Monica P. Colaiácovo[2], Hyun-Min Kim[1]*

**1** School of Pharmaceutical Science and Technology, Tianjin University, Tianjin, China, **2** Department of Genetics, Harvard Medical School, Boston, Massachusetts, United States of America

* hkim@tju.edu.cn

**Data Availability Statement:** All relevant data are within the manuscript and its Supporting Information files.

## Abstract

Histone methylation is dynamically regulated to shape the epigenome and adjust central nuclear processes including transcription, cell cycle control and DNA repair. Lysine-specific histone demethylase 2 (LSD2) has been implicated in multiple types of human cancers. However, its functions remain poorly understood. This study investigated the histone demethylase LSD2 homolog AMX-1 in *C. elegans* and uncovered a potential link between H3K4me2 modulation and DNA interstrand crosslink (ICL) repair. AMX-1 is a histone demethylase and mainly localizes to embryonic cells, the mitotic gut and sheath cells. Lack of AMX-1 expression resulted in embryonic lethality, a decreased brood size and disorganized premeiotic tip germline nuclei. Expression of AMX-1 and of the histone H3K4 demethylase SPR-5 is reciprocally up-regulated upon lack of each other and the mutants show increased H3K4me2 levels in the germline, indicating that AMX-1 and SPR-5 regulate H3K4me2 demethylation. Loss of AMX-1 function activates the CHK-1 kinase acting downstream of ATR and leads to the accumulation of RAD-51 foci and increased DNA damage-dependent apoptosis in the germline. AMX-1 is required for the proper expression of mismatch repair component MutL/MLH-1 and sensitivity against ICLs. Interestingly, formation of ICLs lead to ubiquitination-dependent subcellular relocalization of AMX-1. Taken together, our data suggest that AMX-1 functions in ICL repair in the germline.

## Author summary

Epigenetic failures in DNA damage repair have long been implicated in multiple types of human cancers, including colorectal and stomach cancer. Several studies reported histone demethylases' roles in DNA damage repair; however, the mechanisms by which histone demethylases contribute to DNA damage repair are not very clear. Here, we describe a mammalian LSD2 homolog AMX-1 in *C. elegans* and uncover a potential link between H3K4me2 modulation and DNA interstrand crosslink repair. AMX-1 mainly localizes to embryonic cells, the mitotic gut and sheath cells. AMX-1 regulates H3K4me2 demethylation and is necessary for sensitivity against interstrand crosslink damage.

**Funding:** This work was supported by the National Natural Science Foundation of China (NSFC No 31972876, http://nsfc.gov.cn/) award to H.M.K. and a National Institutes of Health grant (R01GM072551,https://www.nih.gov) to M.P.C. The funders had no role in study design, data collection and analysis, decision to publish, or preparation of the manuscript.

**Competing interests:** The authors have declared that no competing interests exist.

## Introduction

A nucleosome consists of a H2A, H2B, H3 and H4 histone octamer wrapped around by DNA and further condensed into chromatin [1]. Histones undergo post-translational modifications at various sites along their tails, such as methylation, acetylation, and phosphorylation [2]. These modifications affect chromatin structure and help orchestrate events such as transcription, DNA damage repair, and meiotic crossover recombination [1]. For example, methylation of lysine four on histone H3 (H3K4) is typically associated with increased transcription (euchromatin), while methylation of K9 on H3 is associated with heterochromatin [3–5].

LSD1, the first histone demethylase identified, reverses mono- or di-methylation of H3K4 in promoter regions in human cells [6]. Studies in flies and yeast revealed impaired fertility in the absence of LSD1 [7–9]. Later, *C. elegans* LSD1/SPR-5 mutant analysis revealed progressive sterility through generations with evidence of a progressive accumulation of H3K4me2 for a subset of genes [10]. Other studies discovered that this histone demethylase is essential for DNA double-strand break (DSB) repair and p53-dependent germ cell apoptosis in the *C. elegans* germline [11], linking H3K4me2 modulation to DSB repair. H3K4me2 levels are important for recovery from DNA damage repair due to global gene expression changes [12]. Also, H3K4me2 is modulated by the Fanconi anemia pathway connecting the regulation of histone methylation level to DNA damage repair [13].

While LSD1 has been investigated extensively by multiple laboratories, another histone demethylase homolog, LSD2, has been relatively less studied. Studies show that LSD2 functions as a histone demethylase and regulates genomic imprinting and gene expression [14,15]. Other studies demonstrate its overexpression in different types of cancer cells; however, we are still far from understanding the mechanisms of how LSD2 works [16–19].

*C. elegans* has three histone demethylase homologs, SPR-5, AMX-1 and LSD-1. Several reports focused on the roles played by the mammalian LSD1 homolog *C. elegans* SPR-5 in DNA damage repair. However, the functions of the mammalian LSD2 homolog, AMX-1, are not as well understood. Based on the high degree of amino acid sequence conservation shared between AMX-1 and SPR-5, and that expression of AMX-1 can compensate for the absence of SPR-5, we hypothesized that AMX-1 is engaged in DNA damage repair. Here we have identified a role for the histone demethylase AMX-1, conserved from worms through humans, in the DNA interstrand crosslink (ICL) repair in *C. elegans* for the first time.

We show that AMX-1 mainly localizes to embryonic, the mitotic gut and sheath cells, suggesting a role in mitotic nuclei. Consistently, lack of AMX-1 expression resulted in embryonic lethality, a decreased brood size and a disorganized premeiotic tip where mitotically dividing nuclei are located in the germline. Loss of AMX-1 function activates the CHK-1 kinase signal downstream of ATR and leads to the accumulation of RAD-51 foci accompanied by a higher level of DNA damage-dependent apoptosis in the germline, suggesting that its function is necessary for DNA damage repair. *amx-1* mutants exhibited reduced MutL/MLH-1 levels and tolerance to ICLs. Furthermore, AMX-1 undergoes relocalization upon induction of ICLs, in a manner dependent on the ubiquitination pathway. Lack of AMX-1 expression leads to a significant increase in H3K4me2 levels in both mitotic and meiotic regions of the germline, suggesting that AMX-1 regulates H3K4me2 modulation. Although AMX-1 and SPR-5 exhibit reciprocal up-regulated expression upon lack of each other, they are not fully functionally redundant. Altogether, our analysis supports a model in which the histone demethylase AMX-1 plays a role in ICL repair.

## Results

### AMX-1 is a conserved histone demethylase

*amx-1* (open reading frame R13G10.2) is one of the three histone demethylases identified based on amino acid sequence homology in *C. elegans*. *amx-1* encodes for an 824 amino acid protein that contains SWIRM and amine oxidase domains (Figs 1 and S1). Unlike SPR-5, which is a homolog of human LSD1 in *C. elegans*, AMX-1 does not contain a TOWER domain and is therefore classified as an LSD2 homolog. Both SWIRM and amine oxidase domains are well conserved between worm and human LSD2 suggesting these are functional orthologs (38% identity and 64% similarity for the SWIRM domain and 37% identity and 73% similarity for the amine oxidase domain). The high degree of conservation of both SWIRM and amine oxidase domains throughout species supports concurrent conservation of the AMX-1 role in different species. However, *C. elegans* AMX-1 does not contain a zinc-finger motif found in its mammalian counterpart, reflecting evolutionary variation as well.

### AMX-1 is required for normal fertility and germline progression

The *amx-1* deletion mutant (*ok659*) carries a 2636 base pair deletion encompassing the entire SWIRM and 89% of the amine oxidase coding region (Figs 1A and 1B and S1). Therefore, it is predicted that the deletion mutation would result in the loss of potential histone demethylase function. As expected, analysis of wild type and *amx-1* mutant lysates by RT-PCR revealed a ~90% reduction of mRNA expression of *amx-1* compared to control, suggesting the lack of mRNA expression in the mutant (Fig 1C).

*amx-1* homozygous mutants do not exhibit larval arrest (% adult) or a high incidence of males (Him, % male) among its progeny suggesting that AMX-1 does not have a role during

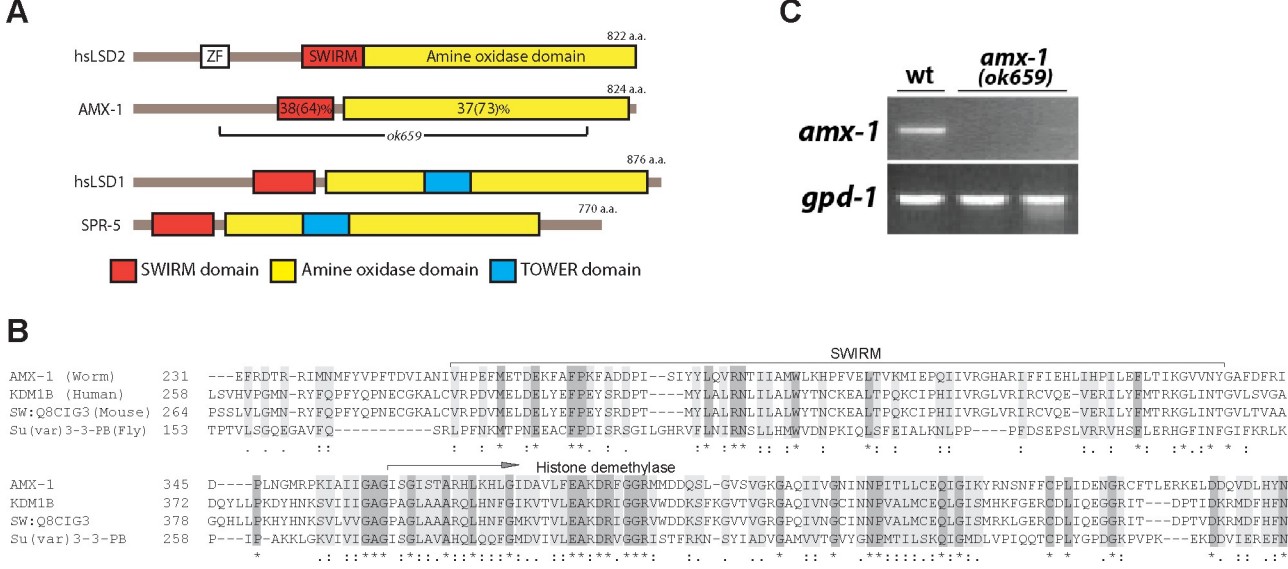

**Fig 1. LSD1/2 homology between *C. elegans* and humans.** (**A**) Schematic of human LSD1 and LSD2 proteins shown with the corresponding *C. elegans* homologs underneath. *amx-1(ok659)* carries a large deletion encompassing the SWIRM domain and most of the amine oxidase domain. ZF = zinc finger domain. (**B**) Representative amino acid sequence alignment of worm, human, mouse and fly LSD2 homologs. Conservation of the SWIRM and amine oxidase domains are shown. For a full amino acid sequence alignment, see S1 Fig. (\*) denotes a single, fully conserved residue; (:) represents conservation between groups of highly similar properties and (.) indicates conservation between groups of weakly similar properties. (**C**) Semi-quantitative RT-PCR analysis of whole worm lysates from *amx-1(ok659)* mutants reveals a 90% reduction in mRNA expression compared to wild type, suggesting that *ok659* is a null mutant. *gpd-1* is an internal mRNA expression control.

larval development or sex chromosome segregation ([Fig 2A]). However, homozygous mutants exhibit a significant 2.4-fold increase in embryonic lethality (2.7% compared to 1.1%, P = 0.0004 by the two-tailed Mann-Whitney test, 95% C.I.) and 17% reduction in brood size compared to wild type (256 compared to 309, P = 0.0001 by the two-tailed Mann-Whitney test, 95% C.I.). Also, the brood size was not significantly reduced in the heterozygotes, suggesting that *ok659* is a recessive allele ([S2 Fig]).

Embryonic lethality and reduced fertility suggest a role for AMX-1 in promoting meiotic chromosome segregation. Therefore, we investigated whether AMX-1 plays a role in the germline by assessing germ cell progression in DAPI-stained gonads from age-matched hermaphrodites ([Fig 2B]). We observed a decrease in the length of the premeiotic tip, which is occupied by mitotic nuclei, in *amx-1* mutants compared to wild type (26% reduction, 72.2 μm for wild type and 53.5 μm for *amx-1*). Conversely, neither the length of the transition zone (leptotene/zygotene stages), where meiosis initiates (65.8 μm for wild type and 68.4 for *amx-1*), or the overall gonad length were altered (from distal tip to late pachytene, 358.3 μm for wt and 338.5 for *amx-1*, P = 0.2565). Furthermore, we observed an increased number of gonads with gaps in *amx-1* mutants while nuclei were organized in a normal temporal and spatial manner in wild

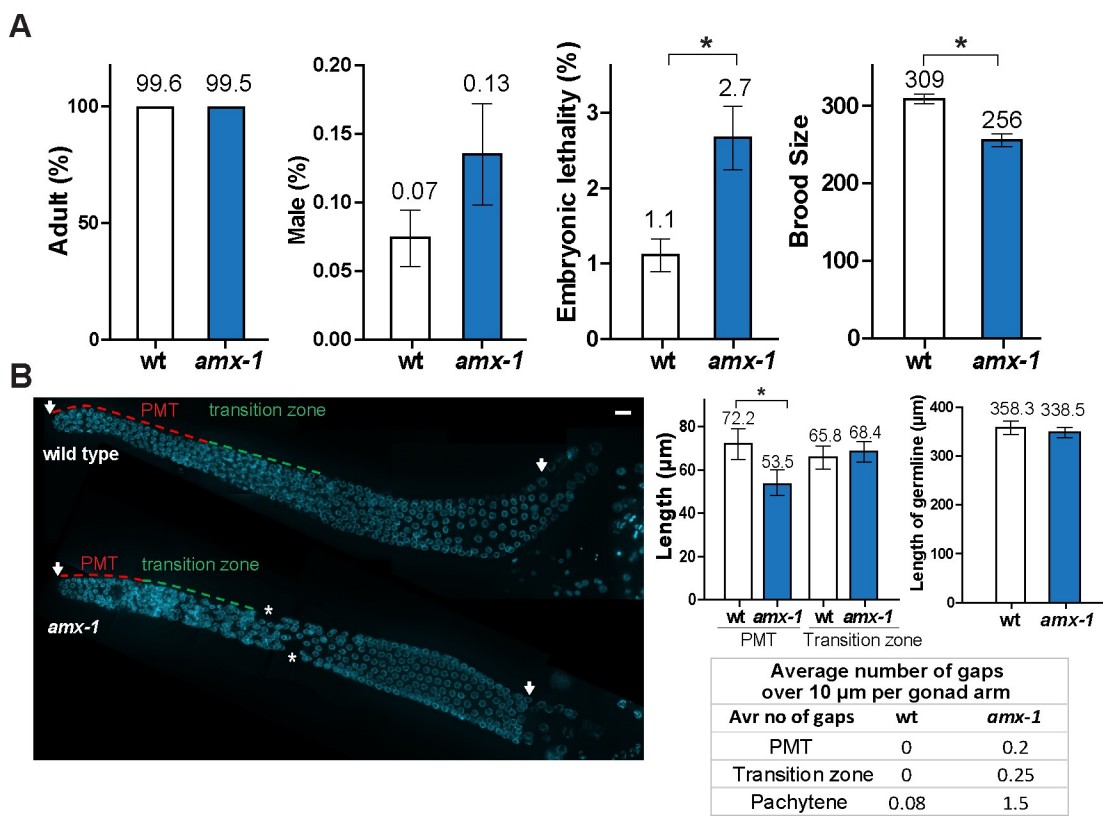

**Fig 2. AMX-1 is a conserved protein required for proper embryonic survival, brood size and germ cell progression. (A)** Plate phenotypes of *amx-1* mutants. % adult to identify potential larval arrest, % male, embryonic lethality and brood size are scored for the progeny of worms of the indicated genotypes. Error bars represent standard error of the mean. N = 24 for each genotype. *P<0.0001 by the two-tailed Mann-Whitney test, 95% C.I. **(B)** Left, Images show DAPI-stained nuclei (blue) in germlines from wild type and *amx-1* mutants. The length of the premeiotic tip (PMT) in the *amx-1* mutant is shorter than in wild type, while neither the length of transition zone or the total length of the germline are affected. The gonads in *amx-1* mutants also displayed an increase in gaps (asterisks) compared to wild type. Bar, 10μm. Right, Quantification of the lengths of the PMT (red dash line), transition zone (green dash line), and germline (indicated by arrows), and the number of gaps (asterisks) per gonad for the indicated genotypes. Error bars indicate standard deviation. N = > 24 gonads.

type gonads. Taken together, these results suggest AMX-1 is required for proper germ cell progression with a primary role in mitotic progression.

## AMX-1 localizes to embryonic cells and mitotic nuclei

To gain some insight into the function of AMX-1, *amx-1*::GFP transgenic animals were generated using CRISPR-Cas9 as described in [20,21]. *amx-1*::GFP transgenic animals exhibit a normal brood size while *amx-1* mutants exhibit a decreased brood size suggesting that *amx-1*:: GFP transgenic animals express functional AMX-1 (P = 0.7922, S3 Fig).

We examined *amx-1*::GFP localization by immunostaining dissected hermaphrodite gonads with an anti-GFP antibody. Consistent with the phenotypic results, AMX-1-GFP signal is mainly observed in the nuclei of gut cells, embryonic cells (regardless of their stage), and sheath cells (Fig 3A and S1 Table). No obvious signal was observed in the control (N2) strain or upon *amx-1* RNAi depletion, which supports the specificity of the antigen-antibody reaction (Figs 3A and S4). Moreover, nuclei from premeiotic tip to diakinesis stages for the most part do not exhibit AMX-1-GFP signal, except for 5% of the premeiotic tip and pachytene nuclei where it was detected along the DAPI-stained chromosomes (S5 Fig). AMX-1-GFP signal was not detected either in sperm or pharyngeal neurons (Fig 3B).

*amx-1* gene expression had been previously reported to be up-regulated in *spr-5* mutants based on microarray studies [10,11]. To further understand the interplay of these two genes at the cellular level, we examined the expression of AMX-1 in the *spr-5* mutants. Interestingly, lack of SPR-5 expression induces a further increase in AMX-1 levels in gut and embryonic cells (4.9- and 1.9-fold induction, respectively, Fig 3A), however, no evident induction from premeiotic tip to diakinesis or mitotic sheath cells was observed. These observations suggest tissue-specific compensation involving the two histone demethylase genes.

## Lack of either AMX-1 or SPR-5 results in reciprocal up-regulated expression in mitotic cells

To further investigate the interplay between the two histone demethylases, AMX-1 and SPR-5, we examined their expression levels using quantitative real-time PCR (Fig 3C). While the lack of *amx-1* leads to ~3.2 fold up-regulation of *spr-5* mRNA expression, the absence of *spr-5* results in ~5.8 fold up-regulation of *amx-1* suggesting a reciprocal compensation between *amx-1* and *spr-5* expression. Consistent with our immunostaining results, this mRNA expression pattern suggests that AMX-1 expressed in mitotic cells might compensate, at least in part, for the lack of SPR-5. Altogether, these observations support AMX-1 expression in embryonic cells and mitotic nuclei in line with its mitotic roles observed in our phenotypic analysis (Fig 2).

## AMX-1 regulates H3K4me2 methylation with properties distinct from SPR-5

Given that AMX-1 contains a conserved amine oxidase domain and is up-regulated upon lack of SPR-5 expression, we next examined AMX-1 histone demethylase activity *in vivo* by staining for dimethylated H3K4 level. Specifically, dissected gonads from wild type, *spr-5* and *amx-1* age-matched adult hermaphrodites were probed for H3K4me2. Consistent with previous reports, *spr-5* mutants display a significant increase in H3K4me2 signal compared to wild type in embryos and at the pachytene stage (Fig 4, P = 0.0001, two-tailed Mann-Whitney test, 95% C.I., [10,11]). Lack of AMX-1 expression results in a significant increase in H3K4me2 level in both embryos and germlines. We found that the signal intensities are significantly increased in

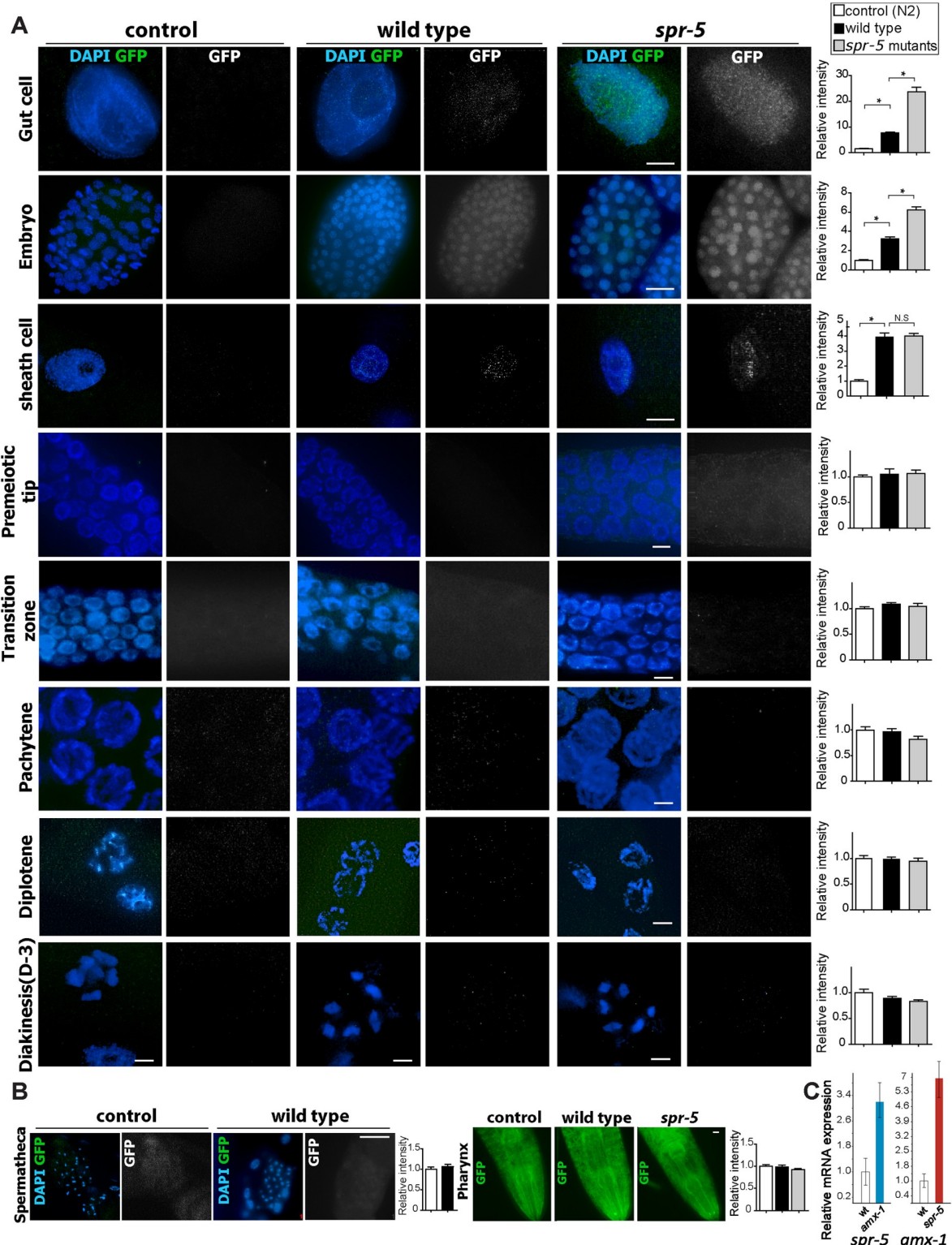

**Fig 3. AMX-1 is expressed in the nuclei of gut, embryonic and sheath cells and is further upregulated in absence of SPR-5. (A)** Immunolocalization of AMX-1-GFP utilizing an anti-GFP antibody in the indicated mitotic cells and at different stages in the gonads of control (N2 worms), wild type (*amx-1::GFP* worms) and *spr-5; amx-1::GFP* worms. AMX-1-GFP signal is detected in the nuclei of mitotic cells including gut, embryonic and sheath cells. The absence of SPR-5 expression results in increased AMX-1 signal in mitotic cells. Right, relative intensity of AMX-1 expression measurements. *P<0.0001 by the two-tailed Mann–Whitney test, 95% C.I. No distinct signal is

detected in either *amx-1::GFP* or *spr-5;amx-1::GFP* worms during meiotic progression (Transition zone, Pachytene, Diplotene and Diakinesis). Bars, 10μm. **(B)** At spermatheca or pharynx, no distinct signal is detected in either *amx-1::GFP* or *spr-5;amx-1::GFP* worms. Bars, 10 μm. **(C)** mRNA expression levels of *spr-5* and *amx-1* quantified by qRT-PCR. Relative mRNA expression measured in wild type, *amx-1*, and *spr-5* mutants. ~3.2-fold induction of *spr-5* mRNA in *amx-1* and ~5.8-fold induction of *amx-1* mRNA expression in *spr-5* mutants are detected.

embryonic cells (~1.8 fold), premeiotic tip nuclei (~1.8 fold), and pachytene nuclei (~1.8 fold) compared to wild type, supporting a role for *amx-1* in germline and somatic H3K4me2 regulation (P<0.0001 for all three comparisons, two-tailed Mann-Whitney test, 95% C.I.). Interestingly, *amx-1* mutants altered histone methylation level in the premeiotic tip while *spr-5* did not, suggesting that AMX-1 has distinct properties from SPR-5 in modulating histone methylation. Consistent with this, *spr-5;amx-1* double mutants did not display additive or synergistic histone methylation levels relative to *amx-1* single mutants (in premeiotic tip nuclei, P = 0.1461; in embryos, P = 0.3080; in pachytene nuclei, P = 0.6434; and in gut cells, P = 0.7680).

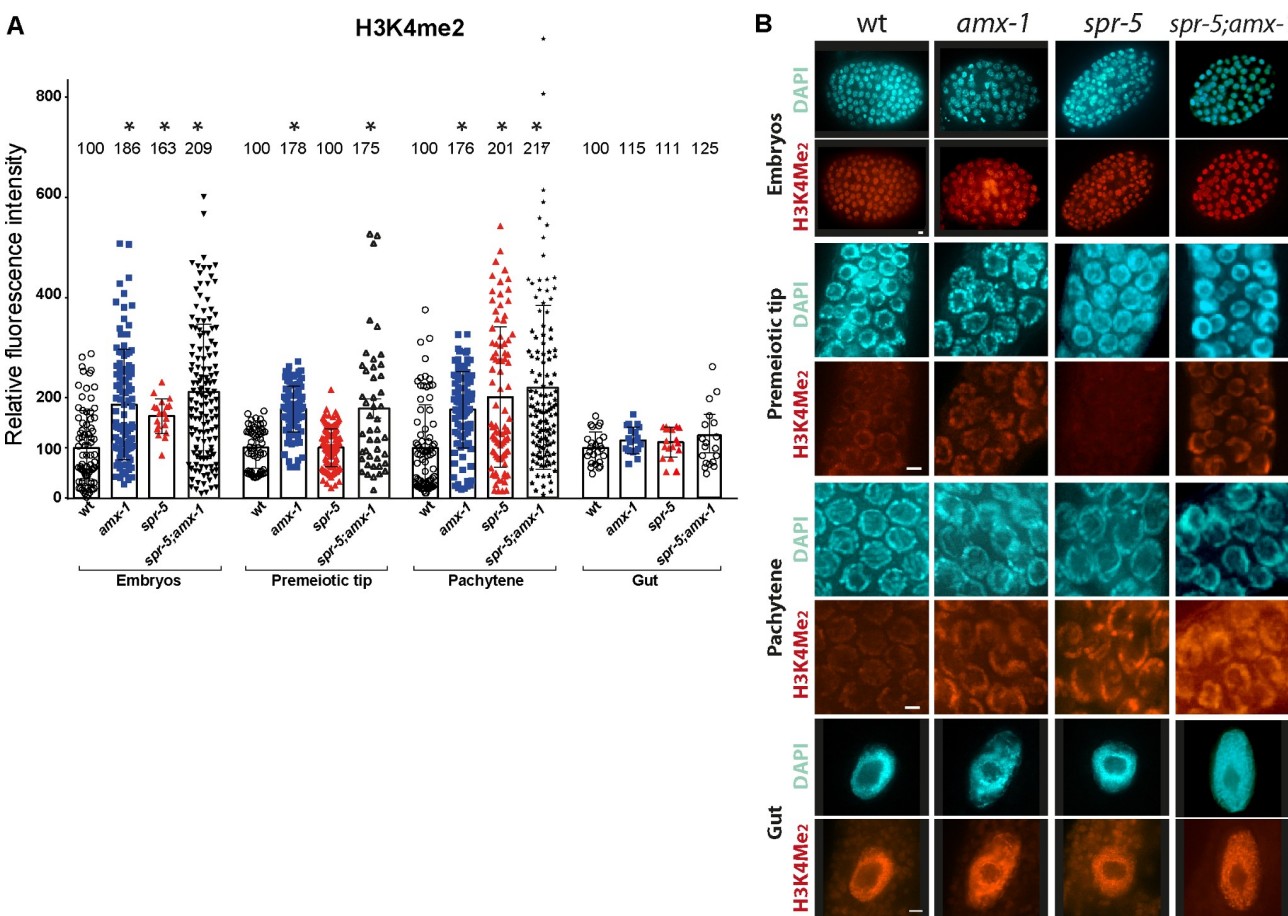

**Fig 4. Lack of AMX-1 expression results in a significant increase in H3K4me2 level in both embryos and germlines. (A)** H3K4me2 signal intensity was significantly increased in nuclei in embryonic cells (~1.8-fold), premeiotic tip (~1.8-fold), and pachytene (~1.8-fold) compared to wild type (P<0.0001 for all three comparisons). No significant induction was detected in gut nuclei (P = 0.0847 for *amx-1* and wt). *spr-5;amx-1* double mutants did not display additive or synergistic effects on histone methylation levels compared to *amx-1* single mutants (in PMT, P = 0.1461; in embryo, P = 0.3080; in pachytene, P = 0.6434; in gut cells, P = 0.7680 for *amx-1* and *spr-5;amx*). Individual nuclei were sampled for fluorescence intensity using ImageJ. The bars represent the mean fluorescence intensity. N = 5–8 gonads. **(B)** Representative images used for the analysis. Bars, 2 μm. Wider field images are available in S10 Fig.

### The DNA damage checkpoint is activated in *amx-1* germline nuclei

Genotoxic stress or DNA replication stress can lead to checkpoint responses in the *C. elegans* germline. Activation of the mitotic DNA damage checkpoint leads to elevation of ATL-1 (ATR homolog) and phosphorylated CHK-1 (pCHK-1) in germline nuclei [22]. In *amx-1* mutants, elevated levels of pCHK-1 foci are observed in germline nuclei even without exogenous DNA damage exposure (Fig 5A and S2 Table). Increased pCHK-1 signal is detected in embryos, pre-meiotic tip-transition zone and pachytene nuclei and this increase requires the upstream pathway component ATL-1 (*C. elegans* ATR homolog). Consistently, *amx-1* mutants exhibit a higher level of *egl-1* expression, a DNA damage response marker, suggesting the lack of AMX-1 expression activates the DNA damage checkpoint in an ATL-1 dependent manner.

### AMX-1 is required for sensitivity upon interstrand crosslink DNA damage

To examine the role of AMX-1 in DNA damage repair, adult hermaphrodites were exposed to multiple types of DNA damage and embryonic lethality was monitored as described in [23–25] (Fig 5B). As previously shown, lack of SPR-5 induced sensitivity against DSBs by γ-irradiation (IR) exposure [11], however *amx-1* mutants did not alter hatching frequencies compared to wild type, suggesting that AMX-1 function may not be required for DSB repair (Fig 5B, P > 0.9999 at 60 gray dose and P = 0.5072 at 120 gray doses for wt and *amx-1*). Interestingly, exposure to nitrogen mustard (HN2), which produces ICLs, improved hatching levels in *amx-1* mutants compared to wild type (76% vs 94% at 100 μM P = 0.0139 and 51% vs 89% at 150 μM P = 0.0003, while lack of HIM-18, an ortholog of MUS312/Slx4 that is required for ICL repair [23], confers sensitivity as shown by relative hatching of 21% at 100 μM and 26% at 150 μM). Consistently, *amx-1* mutants also exhibited tolerance to cisplatin, which produces DNA crosslinks (83% vs 92% at 300 ug/ml P = 0.0119). In addition to tolerance to nitrogen mustard, *amx-1* mutants displayed chromosome fragments at the premeiotic tip, further supporting a role for AMX-1 in ICL repair (Fig 5C). Neither ICL resistance or sensitivity was observed with lack of SPR-5 expression (S6 Fig) suggesting that AMX-1 and SPR-5 exert separate functions in DNA damage repair.

In addition, UVC, which induces cyclobutane pyrimidine dimers, and camptothecin, which results in a single-ended DNA DSB when collision of a replication fork occurs at the lesion, did not reduce hatching levels in *amx-1* mutants compared to wild type (Fig 5B, for UVC, 99% (wt) vs 100% (*amx-1*) at 100 J/m$^2$, P = 0.166,1 and 97% (wt) vs 96% (*amx-1*) at 200 J/m$^2$, P = 0.6858; for camptothecin, 98% (wt) vs 99% (*amx-1*) at 100 nM, P = 0.4038, 97% (wt) vs 96% (*amx-1*) at 500 nM, P = 0.1374, and 92% (wt) vs 93% (*amx-1*) at 750 nM, P = 0.5669). Exposure to hydroxyurea (HU), which results in a checkpoint-dependent cell cycle arrest, led to no significant changes in hatching in *amx-1* mutants compared to wild type (99% and 99%, respectively, at 15 mM, P = 0.6538, and 93% (wt) vs 97% (*amx-1*) at 25 mM, P = 0.2228) while lack of ZTF-8, an ortholog of RHINO, induced sensitivity as previously shown [22]. Taken together, these results indicate that the function of AMX-1 in DNA repair exhibits a high degree of DNA damage specificity, being required for sensitivity against interstrand crosslinks.

### AMX-1 endows proper expression of MLH-1, thus allowing sensitivity against DNA interstrand crosslink damage

Given that AMX-1 exhibits histone demethylase activity and regulation of histone methylation has been implicated in the regulation of gene expression levels [14,26], we hypothesize that AMX-1 may regulate the expression of DNA repair genes leading to resistance against ICLs rather than directly being engaged in the repair process. To test this, we examined whether

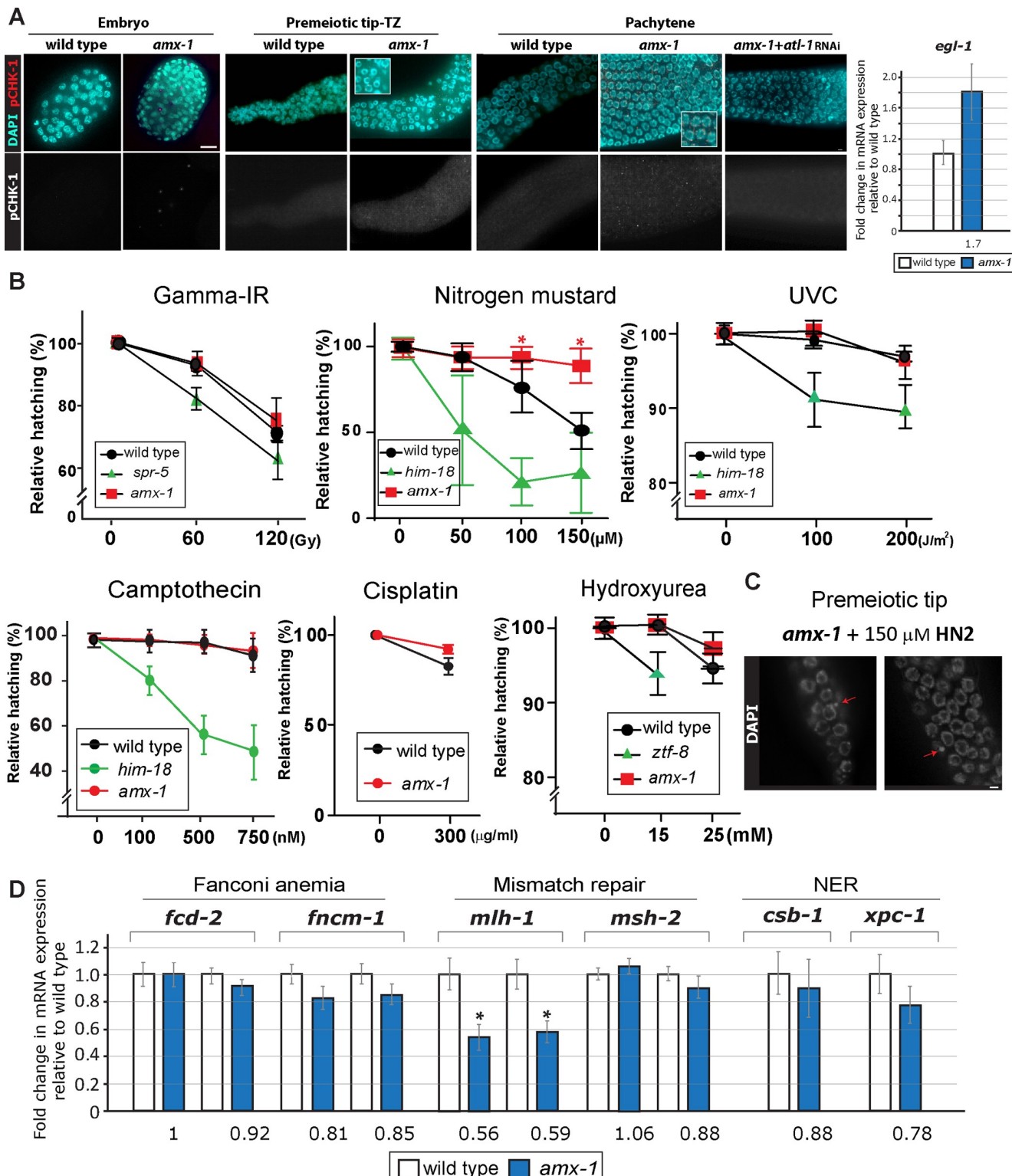

**Fig 5. Absence of AMX-1 induces DNA damage checkpoint activation and AMX-1 is required for sensitivity upon interstrand crosslink damage. (A)** The DNA damage checkpoint is activated in *amx-1* mutants, as shown by the increased pCHK-1 signal observed in embryonic cells, premeiotic tip-transition zone (TZ) and pachytene nuclei. Insets show higher magnification images. Bars, 2 μm. Right, in line with increased pCHK-1 signal, *amx-1* mutants exhibit a higher level of *egl-1* mRNA expression. **(B)** Relative hatching of wild type, *amx-1*, *spr-5*, *him-18*, or *ztf-8* mutants after treatment with the indicated doses of gamma-irradiation (IR), nitrogen mustard (HN2), UVC, camptothecin, and hydroxyurea (HU). *spr-5*, *him-18* or *ztf-8* were used as

controls. Exposure to nitrogen mustard improved hatching levels in *amx-1* mutants. Asterisks indicate statistical significance between wild type and *amx-1* calculated by the two-tailed Mann-Whitney test, 95% C.I. **(C)** *amx-1* mutants are susceptible to ICLs. Nitrogen mustard (HN2) exposure leads to increased numbers of gonads with chromosome fragments (arrows) compared to wild type. Representative images of nuclei in the premeiotic tip from *amx-1* mutants treated with 150 μM HN2. 6 out of 30 (20%) gonads contained broken chromosomes for *amx-1* and 0 out of 20 (0%) gonads for wild type. Bar, 2 μm. **(D)** Lack of *amx-1* expression decreases MutL/*mlh-1* expression. Changes of relative mRNA expression levels of Fanconi anemia (*fcd-2* and *fncm-1*), mismatch repair (*mlh-1* and *msh-2)*, and nucleotide excision repair (*csb-1* and *xpc-1*) genes. mRNA expression was monitored with 1–2 sets of qPCR primers spanning exon-exon junctions for each indicated gene. The data shown are the means with SEM analyzed employing an unpaired, two-tailed t-test. Assay repeated three times with at least two independent sample preparations. Tubulin encoding *tba-1* was used as an endogenous reference. Asterisks (*) indicate significant differences in mRNA expression compared to wild type.

AMX-1 regulates the proper expression of genes in the Fanconi anemia (FA), mismatch repair, and nucleotide excision repair (NER) pathways, all of which have been implicated in ICL repair [12,13,27–30]. While mRNA expression of two key FA components, *fcd-2* and *fncm-1*, and two key NER components, *csb-1* and *xpc-1*, was not significantly altered, interestingly, *amx-1* mutants exhibited an ~43% reduction in *mlh-1* expression compared to wild type (Fig 5D). However, no significant change in *msh-2* expression was observed, implying that AMX-1 regulation is constrained to the MutL complex only.

To explore whether proper expression of MLH-1 allows susceptibility to ICLs, we depleted MLH-1 expression by RNAi and assessed the frequency of hatching upon exposure to DNA crosslinking damage by either HN2 or cisplatin. Lack of MLH-1 expression resulted in increased tolerance to both HN2 and cisplatin compared to control, indicating that down-regulation of MLH-1 may be sufficient for the ICL tolerance displayed by *amx-1* mutants (S7 Fig).

## The localization of AMX-1 is altered in response to interstrand crosslink DNA damage in a manner dependent on ubiquitination

To determine whether AMX-1 localization is altered in response to ICL formation, we exposed wild type worms to HN2 and monitored AMX-1 localization. Unlike the unexposed worms, in which AMX-1 signal is present evenly in all of the embryonic and intestine cells, an uneven/mosaic distribution of AMX-1 signal is observed upon HN2 treatment (Fig 6A). Embryos containing both stronger and weaker signals are observed. Overall, average signal intensity is reduced by 30% compared to control upon HN2 treatment (Fig 6B). Also, 5% of embryos and 2% of intestine cells exhibited complete lack of AMX-1 signal following HN2 treatment whereas AMX-1 signal was always observed in untreated control (S1 Table).

Given the altered signal observed in the +HN2 worms together with a predicted ubiquitination site (Lysine 11) from *in silico* analysis [31], these results suggest that AMX-1 may either be relocalized or degraded after ICL DNA damage. To test this idea, we depleted ubiquitin-activating enzyme E1 UBA-1 expression and monitored AMX-1 localization. Depletion of UBA-1 expression with RNAi feeding rescues the uneven or absent AMX-1 signal, suggesting that poly-ubiquitination of AMX-1 leads to relocalization of AMX-1 upon ICL repair (P = 0.0001, comparing with and without UBA-1 depletion, Fig 6B and 6C). Lack of alteration in *amx-1* mRNA expression level upon HN2 exposure further supports post-translational modification (Fig 6D).

## AMX-1 is required for proper DNA damage repair progression in germline nuclei

Nuclei are positioned in a temporal-spatial manner along the germline in *C. elegans*, proceeding in a distal to proximal orientation from mitosis into the various stages of meiotic prophase I. Taking advantage of this well-defined organization, levels of RAD-51 foci, which mark sites undergoing DSB repair [32], were quantitated both in mitotic (premeiotic tip) and meiotic nuclei (transition zone and pachytene) and compared between wild type and *amx-1* mutants (Fig 7A).

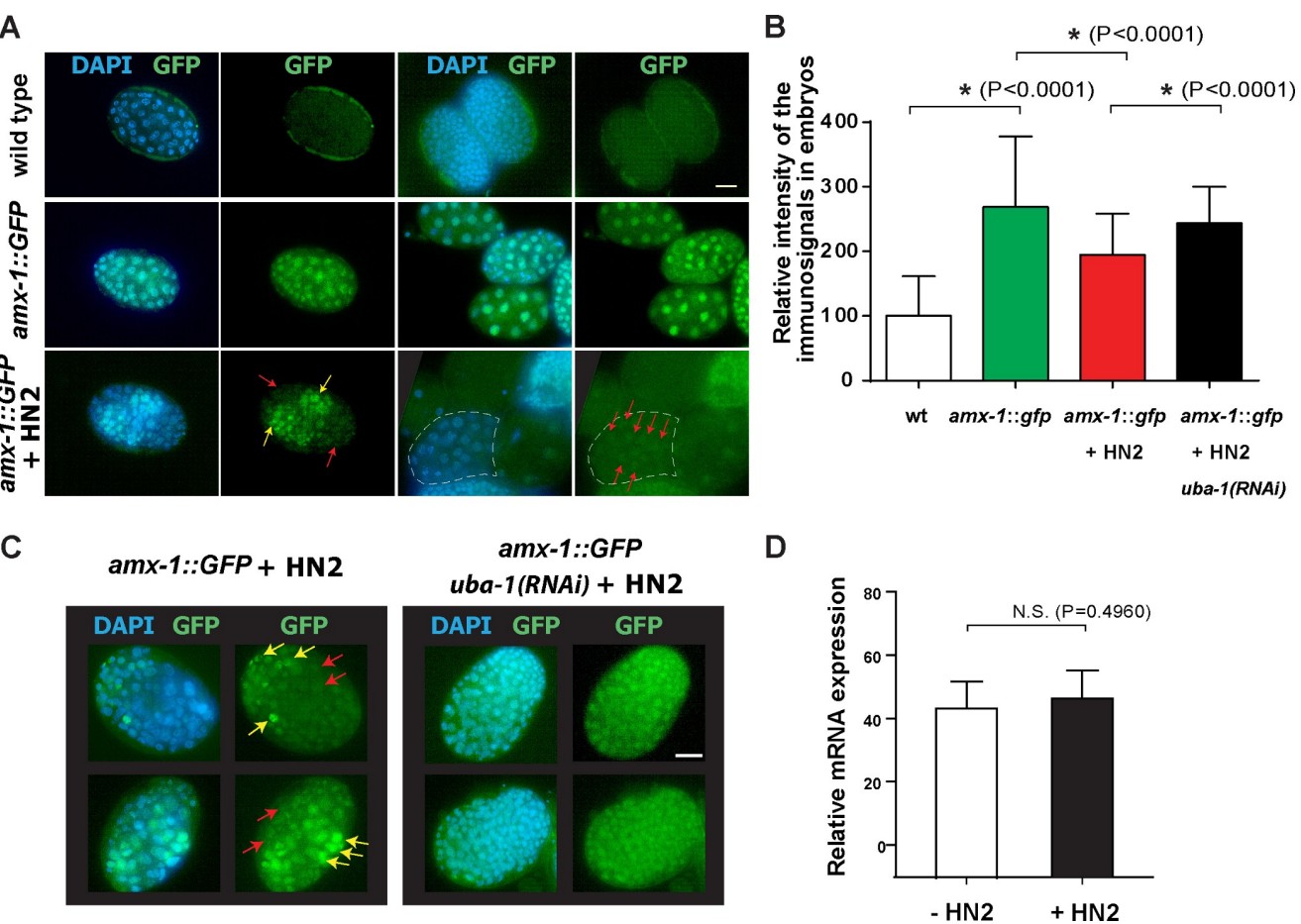

**Fig 6. Interstrand crosslink DNA damage leads to ubiquitination-dependent AMX-1 relocalization. (A)** Nitrogen mustard (HN2) exposure induces relocalization of AMX-1::GFP signals. Arrows indicate embryo cells containing no/weak (red) or stronger (yellow) signals. Bar,10μm. **(B)** Quantitation of relative intensity of the AMX-1::GFP signal in embryos from the indicated genotypes. Asterisks indicate statistical significance. 66–250 embryos from 6–8 worms were analyzed. Error bars represent standard deviations. **(C)** Depletion by RNAi of the E1 ubiquitin-activating enzyme UBA-1, suppresses the uneven/mosaic signals exhibited in *amx-1*::GFP transgenic animals. AMX-1::GFP signal intensity from embryos are quantitated in panel B. P = 0.1084 from *amx-1*::*GFP* + HN2. **(D)** mRNA expression level of *amx-1* is not altered after HN2 treatment. P = 0.4960. Error bars represent standard deviations.

In wild type, only a few RAD-51 foci are detected in mitotic nuclei in the premeiotic tip of the germline and these are mainly derived from single stranded DNA gaps formed at stalled replication forks or resected DSBs resulting from collapsed replication forks [33]. During meiotic prophase, SPO-11-dependent programmed meiotic DSBs are induced and higher numbers of RAD-51 foci are detected peaking in mid-pachytene. In *amx-1* mutants, the mildly elevated levels of RAD-51 foci observed at the premeiotic tip were not significantly different compared to wild type (Fig 7B, an average of 0.1 and 0.2 RAD-51 foci/nucleus for wild type and *amx-1*, respectively). This mild elevation in the levels of RAD-51 foci persisted from mitosis through the mid pachytene stage of meiosis, but in late pachytene the levels of RAD-51 foci were significantly higher in *amx-1* mutants (an average of 3.2, 5.2, 4.1 RAD-51 foci/nucleus were observed in *amx-1* compared to 3.1, 4.4, 3.2 for wild type at early, mid and late pachytene, respectively; P = 0.8059, P = 0.0743 and P = 0.0041(*) for each stage). Nuclei containing higher numbers of RAD-51 foci were more frequently observed in late pachytene in *amx-1* mutants compared to wild type (>10 RAD-51 foci/nucleus in 10/25 *amx-1* gonads compared to 0/25 gonads in wild type (Fig 7C). Interestingly, the higher levels of RAD-51 foci detected by late pachytene were no

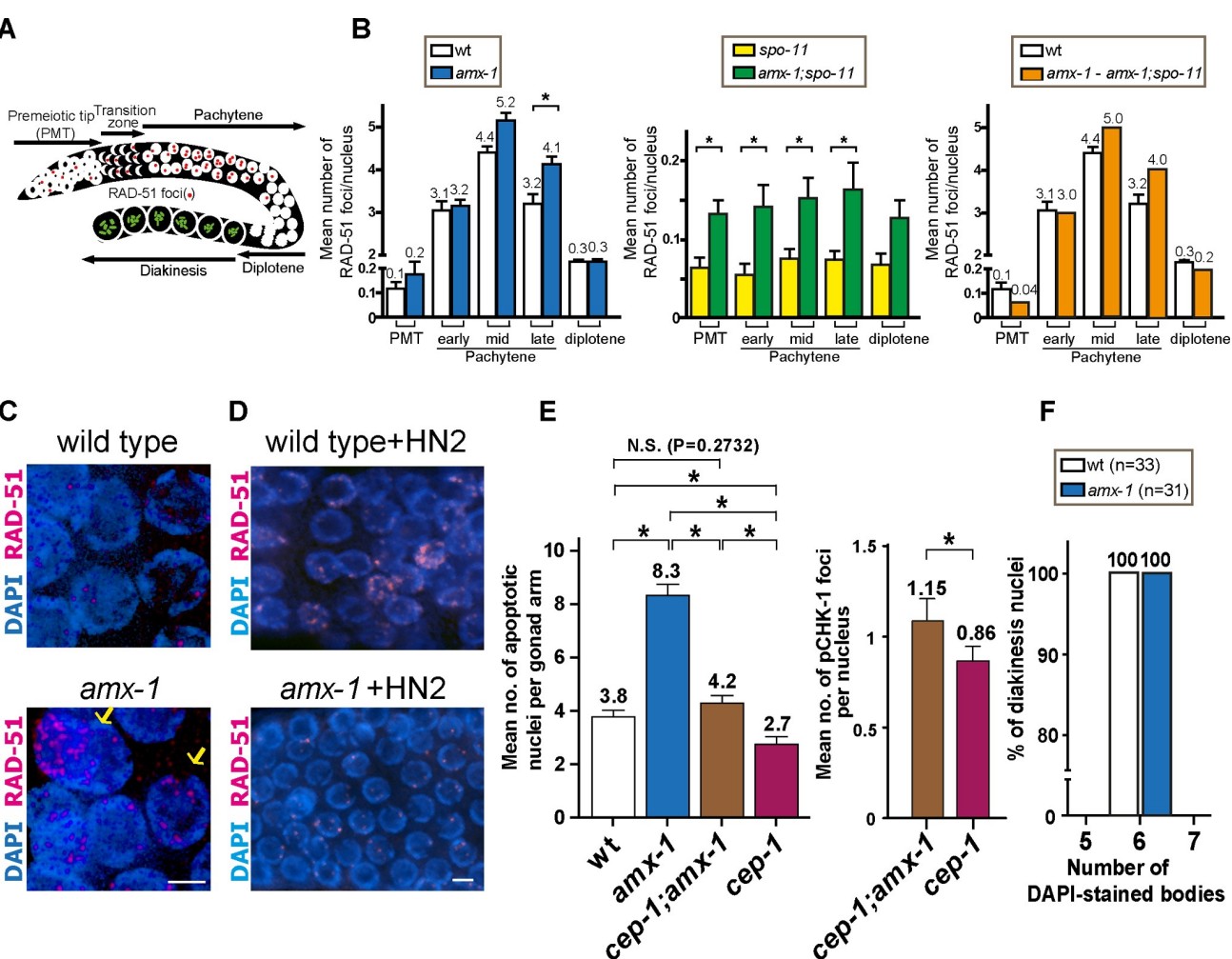

**Fig 7. AMX-1 is required for DNA repair and lack of AMX-1 activates *cep-1*/p53-dependent germ cell apoptosis.** (A) Diagram of a *C. elegans* germline indicating the germline nuclei progression and the position of zones scored for RAD-51 foci. Image is modified from Kim *et al* [22]. (B) Histogram represents the quantitation of RAD-51 foci in germlines of the indicated genotypes. Shown are the mean numbers of RAD-51 foci observed per nucleus on each zone along the germline axis (x-axis). We compared the levels of RAD-51 foci between wild type and *amx-1* mutants (left) and monitored SPO-11-independent (middle) and SPO-11-dependent (right panel) RAD-51 foci levels. To identify the levels of SPO-11-dependent (meiotic) foci, the mean number of RAD-51 foci observed in *amx-1;spo-11* mutants was subtracted from those in *amx-1* mutants. Error bars represent standard error of the mean. Asterisks indicate statistical significance by the two-tailed Mann–Whitney test, 95% C.I. (C) *amx-1* mutants contain pachytene nuclei with >10 RAD-51 foci per nucleus which may undergo apoptosis (indicated by yellow arrows, 10 out of 25 gonads). Bar, 2 μm. (D) High-resolution images of pachytene nuclei from wild type and *amx-1* mutants with 150 μM HN2 treatment immunostained for RAD-51. Bar, 2 μm. (E) Left, quantification of germline apoptosis in the indicated genotypes. *cep-1*/p53 is a mutant with defective DNA damage-induced apoptosis utilized as a control. Asterisks indicate statistical significance. P = 0.2732 for wt and *cep-1;amx-1*. P < 0.0001 all others, by the two-tailed Mann–Whitney test, 95% C. I. Right, Quantification of pCHK-1 foci in pachytene nuclei in the indicated genotypes. P = 0.0299 for *cep-1;amx-1* and *cep-1* by the t-test, 95% C.I. (F) Number of DAPI-stained bodies observed in diakinesis oocytes from the indicated genotypes. The number of -1 oocytes scored (n) is indicated next to the genotypes.

longer observed at the diplotene stage (0.3 foci/nucleus for both genotypes, P = 0.1500) suggesting completion of DNA damage repair upon exit from late pachytene and entrance into the diplotene stage. However, repair of the elevated levels of RAD-51 foci detected in late pachytene may not be dependent on non-homologous end-joining since CKU-80 depletion in *amx-1* mutants did not result in elevated levels of RAD-51 foci persisting into late prophase I (S9 Fig).

To better understand DSB formation in *amx-1* mutants, we monitored the levels of RAD-51 foci in the germlines of *amx-1;spo-11* double mutants which lack the formation of

programmed meiotic DSBs (Fig 7B). Elevated levels of RAD-51 foci in both mitotic (premeiotic tip; PMT) and meiotic (pachytene) nuclei in *amx-1;spo-11* double mutants compared to *spo-11* single mutants suggest these are SPO-11-independent RAD-51 foci. However, when we subtracted the mitotic DSBs found in *amx-1;spo-11* from the total number of RAD-51 foci in *amx-1* mutants, higher levels of RAD-51 were detected in mid and late pachytene in *amx-1* mutants suggesting that SPO-11-dependent meiotic DSBs are also elevated in *amx-1* mutants.

Finally, since *amx-1* mutants exhibit tolerance against HN2, it is plausible that RAD-51 foci levels might decrease in *amx-1* mutants compared with wild-type after HN2 treatment. Indeed, wild type worms exhibited gonads with higher levels of RAD-51 compared to *amx-1* mutants, further validating the ICL tolerance (Fig 7D). Of note, no distinct RAD-51 foci are observed in wild type control embryos after HN2 exposure (S8 Fig).

Taken together, in line with our phenotypic analysis, the pattern of elevated RAD-51 foci suggests unrepaired interstrand crosslink damage accumulated from mitotic nuclei transduced and carried over into meiosis in *amx-1* mutants.

### Increased germ cell apoptosis in *amx-1* mutants is CEP-1/p53-dependent

Unrepaired DNA damage can activate a DNA damage checkpoint resulting in increased apoptosis during late pachytene in the *C. elegans* germline [34]. We examined whether the elevated levels of RAD-51 foci observed in *amx-1* mutants induced a high level of apoptosis. In *amx-1* mutants, increased levels of germ cell apoptosis were observed compared to wild type (8.3 (*amx-1*) compared to 3.8 (wt) apoptotic nuclei; $P < 0.0001$ the two-tailed Mann-Whitney test, 95% C.I., Fig 7E). We further tested whether the high level of apoptosis was mediated by the CEP-1/p53 DNA damage checkpoint. Interestingly, the elevated germ cell apoptosis in *amx-1* mutants was suppressed by 2-fold in *cep-1/p53; amx-1* double mutants (4.2 for *cep-1;amx-1*, $P < 0.0001$ for *amx-1* and *cep-1;amx-1*) suggesting that the elevated apoptosis stems from the activation of a DNA damage checkpoint in late pachytene as a result of damage incurred in the germline. Consistently, DAPI-stained fragments observed in premeiotic tip nuclei upon HN2 exposure (Fig 5C) were also rescued in *cep-1;amx-1* mutants (6.7% gonads contain fragments vs. 20% in *amx-1*). Interestingly, the *cep-1;amx-1* mutant exhibited a mild but higher level of apoptosis compared to a *cep-1* single mutant (2.7 for *cep-1*, $P < 0.0001$), suggesting that some portion of the observed germ cell apoptosis is independent of the *cep-1/p53* pathway [35]. The higher level of pCHK-1 foci in the double mutant also supports this idea (Fig 7E).

### AMX-1 is not required for regulating meiotic crossover frequency

To determine whether AMX-1 plays a role in meiotic crossover formation, we monitored crossover frequency in *amx-1* mutants compared with wild type. We found six pairs of attached homologous chromosomes were detected in late diakinesis oocytes in *amx-1* mutants, suggesting that crossover formation resulted in the formation of chiasmata at levels similar to wild type (Fig 7F). Consistently, no discernible larval arrest or high incidence of male phenotype supports the idea that AMX-1 does not have a role during larval development or sex chromosome segregation (Fig 2A). Altogether, our data indicate that while AMX-1 is required for proper sensitivity against mitotic ICLs, it is not requisite for interhomolog crossover formation.

## Discussion

### AMX-1 is expressed in mitotic cells

AMX-1 signal is predominantly detected in somatic and mitotic germline nuclei and rarely in pachytene nuclei, suggesting that AMX-1 is generally expressed in mitotic cells, unlike SPR-5

which localizes to both mitotic and meiotic germline nuclei [13]. Consistently, an *in situ* hybridization assay also supports the expression of AMX-1 in mitotic cells [36]. Of note, another report found neuronal expression of mNeonGreen under the *amx-1* promoter in a chromosome integrated plasmid [37]. However, no discernible signal was detected in neuronal regions using our CRISPR-Cas engineered transgenic worms carrying AMX-1::GFP located at the endogenous locus (Fig 3). Thus, the difference is likely owing to an expression of AMX-1 in the nonendogenous plasmid-based genome integration.

## AMX-1 and SPR-5 exhibit reciprocal up-regulated expression in mitotic cells, but are not fully functionally redundant

The compensation of gene expressions between *amx-1* and *spr-5* was evidenced by quantitative real-time PCR (Fig 3C). Consistently, immunostaining results found that AMX-1 expression increased upon lack of SPR-5 in the mitotic embryonic cells and gut cells, where the two histone demethylases overlap (Fig 3A). However, lack of SPR-5 did not lead to elevation of AMX-1 signal in mitotic sheath cells or germline nuclei indicating that these histone demethylases do not execute fully redundant functions and execute tissue-specific roles. In line with this idea, lack of AMX-1 expression resulted in accumulation of H3K4me2 in embryonic cells and germline nuclei (Fig 4). Interestingly, the lack of SPR-5 did not affect H3K4me2 levels in the premeiotic tip, while H3K4me2 levels were elevated at the premeiotic tip in the *amx-1* mutants suggesting a non-redundant function for the two genes. *spr-5;amx-1* double mutants, lacking expression of both histone demethylases, revealed H3K4me2 levels similar to those in *amx-1* single mutants, further supporting this idea. Similarly, no distinct changes were observed in embryonic cells or pachytene nuclei between single and double mutants.

Interestingly, AMX-1::GFP signal was detected in gut cells (Fig 3). Studies in other species have identified roles for histone methylation in maintaining intestine integrity and proper immune response [38,39]. Given that AMX-1 is expressed in the gut cells and further altered by lack of SPR-5, the abovementioned functions may also exist in *C. elegans*.

## AMX-1 endows proper sensitivity against interstrand crosslink DNA damage

AMX-1 and SPR-5 work together to maintain histone methylation levels. On the other hand, they also engage in independent DNA damage repair pathways. While SPR-5 responds to DSBs, AMX-1 endows sensitivity against interstrand crosslink DNA damage and undergoes ubiquitination-dependent relocalization upon ICL induction (Figs 5 and 6). However, *spr-5* mutants display no distinct phenotype upon ICL damage and *amx-1* mutants are not sensitive to exogenous DSBs (Figs 5B and S6). Second, the expression pattern of the two also supports their independent functions. Specifically, the expression of AMX-1 increases upon lack of SPR-5 in gut and embryonic cells (Fig 3). However, AMX-1 signal does not increase in the meiotic region where SPR-5 expression is normally enriched, suggesting that AMX-1 and SPR-5 may possess independent roles. It is interesting to note that although AMX-1 rarely localizes to germline nuclei, lack of AMX-1 leads to increased levels of H3K4 dimethylation in germline nuclei (Fig 4).

Of note, a recent study by Wang *et al.* reported that the *amx-1* mutant is resistant to UV irradiation and UV-induced H3K4me2 is independent of nucleotide excision repair [12]. While this may seem to be contradictory to our result (Fig 5B), our assay exposes adult worms to UVC to measure levels of hatching (embryonic viability) and therefore is focused on effects taking place within the germline and/or embryonic development. Wang *et al.* monitored purely larval growth, which could account for the different DNA damage sensitivity outcomes.

In *C. elegans*, upon response to replication stress, the replication checkpoint proteins ATL-1 (ATR), CHK-1 (CHK1), and RPA-1 are activated and monoubiquitylate interstrand cross-link repair (ICLR) component FCD-2 to its downstream effectors [40–42]. Likewise, *amx-1* mutants exhibit activation of the ATR/CHK-1 pathway and p53/CEP-1-mediated DNA damage-induced apoptosis, which supports the idea that AMX-1 is engaged in ICLR (Figs 7 and 8). However, we cannot exclude the possibility that AMX-1 plays other roles in addition to ICL damage repair as suggested by the residual apoptosis found in *cep-1;amx-1* double mutants compared with *cep-1* single.

## ICL resistance via defective mismatch repair

DNA repair components are necessary for repairing DNA lesions and providing proper DNA damage resistance. These include DNA homologous repair genes including RAD-51, MCM8 and SPR-5 [11,43,44]. On the other hand, deficiency of the mismatch repair pathway involves a wide range of DNA damage tolerance, thereby increasing chemoresistance and tumor heterogeneity [28]. Similarly, the lack of AMX-1 induced DNA interstrand crosslinking tolerance followed by an accumulation of RAD-51 foci (Fig 7). RAD-51 foci mark unrepaired interstrand crosslink damage; on the other hand, up-regulation of RAD-51 may also lead to the HN2 tolerance exhibited by *amx-1* mutants [44,45]. However, given that the *amx-1* mutant does not exhibit tolerance against double-strand breaks (Fig 5B), the drug resistance from RAD-51 overexpression is implausible.

Alternatively, AMX-1 may target mismatch repair (MMR) and moderate its expression level. Thus, lack of *amx-1* confers resistance to ICL as exhibited in MMR-defective conditions

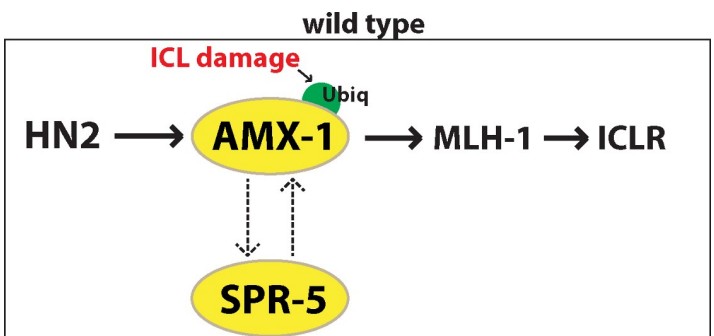

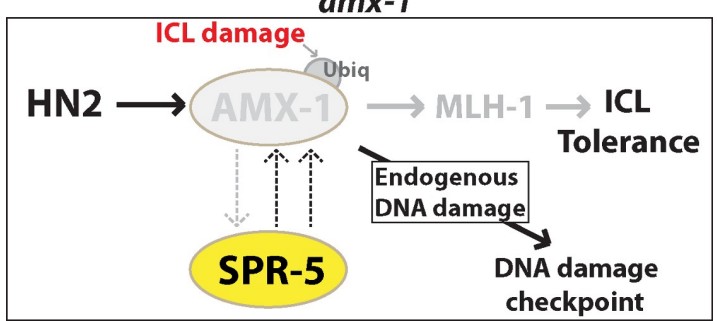

**Fig 8. Model for the role of histone demethylase AMX-1 in DNA damage repair.** Top, in wild-type animals, we propose that AMX-1 and SPR-5 act in both redundant and non-redundant manners. Upon ICL damage, AMX-1 is ubiquitylated and regulates the proper expression of MLH-1 (mismatch repair MutL homolog). Bottom, in the absence of AMX-1, SPR-5 expression is up-regulated. A DNA damage checkpoint is activated in *amx-1* mutants in the absence of exogenous damage, suggesting that lack of AMX-1 responds to endogenous DNA damage. Upon HN2 exposure, lack of AMX-1 leads to reduced MLH-1 expression and ICL tolerance.

[28]. Indeed, reduced expression of the *C. elegans* mutL homolog *mlh-1* in the *amx-1* mutants (Fig 5D) together with the ICL tolerance observed in *mlh-1* depleted worms (S7 Fig), suggest that the tolerance exhibited by *amx-1* mutants likely originates from a defective mutL mismatch repair function.

It is worth noting that although the *amx-1* mutants bear tolerance to ICL damage, they also display chromosome fragmentation at the premeiotic tip (Fig 5C), suggesting that loss of AMX-1 may result in multiple phenotypes not only restricted to defects in ICL repair. Consistent with this, *amx-1* mutants exhibited both CEP-1/p53-dependent and -independent apoptosis (Fig 7E) as well as embryonic lethality, reduced brood size, altered germ cell progression and DNA damage checkpoint activation.

Previously, AMX-1's mammalian homolog LSD2 has been implicated in regulating gene transcription by modulating H3K4me2 [14]. Likewise, since AMX-1 modulates histone methylation level in the germline and somatic cells, identifying the targets for AMX-1 would help understand how it works for repair of ICLs.

## AMX-1 and LSD2

AMX-1 harbors SWIRM and amine oxidase domains found in various species suggesting functional conservation among LSD2 homologs. In contrast, the absence of a zinc-finger motif in AMX-1 reflects its evolutionary variation and diverged function (Fig 1A). Mammalian LSD2 has been reported to act as an E3 ubiquitin ligase and inhibit cancer cell growth by promoting ubiquitination-dependent proteasome degradation [19]. However, the zinc-finger domain, which is essential for E3 ligase activity, was not discovered in AMX-1. Instead, that ubiquitinated AMX-1 is found to be relocalized upon DNA damage suggests functional divergence between the mammalian and *C. elegans* histone demethylases.

Previously, a few histone demethylases have been reported as tumor suppressors. Jumonji C family PHF2 histone demethylase knockdown induced resistance to anticancer agents in a p53-dependent manner [46]. Its abnormal expression in breast, head, and neck cancers suggests the potential function of PHF2 as a tumor suppressor [47,48]. Jumonji D3 is another tumor suppressor induced during differentiation of glioblastoma stem cells and it regulates p53 stabilization [49]. Similarly, increased tolerance against interstrand crosslinking agents in the AMX-1 lacking worms suggests the potential target of AMX-1 as a tumor suppressor. In line with our observations, a few reports found abnormal expression of LSD2 has been implicated in human cancers [19,50]. E3 ubiquitin ligase comprised of two zinc-finger motifs inhibits lung cancer growth by targeting O-GlcNAc transferase for polyubiquitylation. However, given that the absence of zinc-finger motifs in AMX-1 still allows the sensitivity to ICLs, an additional tumor suppressor role may be present in human cells.

In summary, our study revealed that AMX-1 is required for proper fertility, embryonic survival and adequate regulation of H3K4me2 in both mitotic and meiotic nuclei in *C. elegans*. Histone demethylase AMX-1 allows proper sensitivity to interstrand crosslinks by modulating expression of the mismatch repair gene *mlh-1* and undergoing ubiquitination-mediated relocalization upon ICL formation.

The study of the roles of histone demethylases in DNA damage repair has gained much attention since the discovery of a role for SPR-5/humanLSD1 in DNA double-strand break repair [11]. The insights we provide on how AMX-1 works for DNA interstrand crosslink repair and CEP-1/p53-dependent germ cell apoptosis shed light on how histone demethylases operate in different DNA damage repair pathway and, akin to a tumor suppressor gene, may serve as a potential therapeutic target. Our study may expand this theme to the context of a previously unknown link between histone demethylase and DNA damage repair.

## Materials and methods

### Strains and alleles

All *C. elegans* strains were cultured at 20˚C under standard conditions as described and the N2 Bristol strain was used as the wild-type [51]. The following mutations were used in this study: LGI: *spr-5(by101)*, *cep-1(lg12501)*, *hT2[bli-4(e937) let-?(q782) qIs48](I; III)*; LGIII: *amx-1 (ok659)*, *ztf-8(tm2176)*, *qC1[dpy-19(e1259) glp-1(q339) qIs26] (III)*, *amx-1(rj20[amx-1::gfp::ha + loxP unc-119(+) loxP]) unc-119(ed3) III)*, *him-18(tm2181)*, *unc-119(ed3) ruIs32 [pie-1p:: GFP::H2B + unc-119(+)](III)*. The *amx-1* deletion mutant (*ok659*) was provided by the *C. elegans* Gene Knockout Project at the Oklahoma Medical Research Foundation, which was part of the International *C. elegans* Gene Knockout Consortium. *amx-1* deletion mutant (*ok659*) carries a 2636 base pair deletion encompassing the entire SWIRM domain and most (89%) of the amine oxidase domain (Figs 1 and S1). To maintain *amx-1(ok659)* mutants, worms were outcrossed to the wild-type N2 and balanced with qC1 with roller and GFP markers.

### Transgenic worms

The *amx-1(rj20)* transgenic worm was generated with CRISPR-Cas9 technology as described [20] [52]. In brief, we built the repair template by inserting GFP+HA between the last and the stop codon of AMX-1. The *unc-119*(+) sequence presented in the repair template rescued the Unc phenotype of injected animals. The transgenic animal was genotyped and outcrossed with wild type six times.

### Analysis of protein conservation and motifs

AMX-1 alignments were performed using the Uniplot Clustal O multiple sequence alignment tool (www.uniprot.org/help/sequence-alignments)).

### Plate phenotyping

The brood sizes scored correspond to the total number of eggs laid by each worm. The level of embryonic lethality corresponds to the percentage of eggs laid that remained unhatched 24 hours post-laying; the level of larval lethality was the percentage of hatched worms that did not survive to adulthood; the level of males (%Him) was the percentage of adults that were males. Entire progeny for at least 20 worms were analyzed for each genotype at each generation reported. Statistical comparisons between genotypes were performed using the two-tailed Mann-Whitney test, 95% confidence interval (C.I.).

### DNA damage sensitivity test

Young adult homozygous *amx-1* animals were picked from the progeny of *amx-1/qC1* parent animals. To assess for IR sensitivity, animals were treated with 0, 60 or 120 Gy of γ-IR from a $Cs^{137}$ source at a dose rate of 1.8 Gy/min. For HN2 sensitivity, animals were treated with 0, 50, 100 or 150 μM of HN2 (mechlorethamine hydrochloride; Sigma) in M9 buffer containing *E. coli* OP50 with slow shaking in the dark for ~20 hours. UV irradiation treatment was performed utilizing the XL-100 Spectrolinker UVC. Worms were exposed to 0, 100 or 200 $J/m^2$ of UVC and plated to allow recovery for 3 hours. Camptothecin (Sigma) treatment was with doses of 0, 100, 500, or 750 nM. HU sensitivity was assessed by placing animals on seeded NGM plates containing either 0, 15, or 25 mM HU for 20–24 hours. After the DNA damage treatments, animals were washed twice with M9 containing Triton X100 (100 ml/L) [22,25]. Hatching sensitivity was examined in >24 animals ~3 hours after HU treatment. For all other damage sensitivity experiments, over 24 animals were plated, ~7 per plate, and hatching was

assessed for the time period of 20–24 hours following treatment. Each assay was replicated at least twice in independent experiments.

### Quantitative analysis for RAD-51 foci

Quantitative analysis of RAD-51 foci was performed as described in [32]. Between five to eight germlines were scored for each genotype. At least 100 nuclei for each zone for a given genotype were analyzed. Statistical comparisons between genotypes were performed using the two-tailed Mann-Whitney test, 95% confidence interval (C.I.).

To monitor RAD-51 foci after HN2 exposure, wild type N2 and *amx-1* mutants were incubated with 150 μM of HN2 in NGM containing *E. coli* OP50 for 20 hours.

### RNA interference

Feeding RNAi experiments were performed at 20˚C as described [53] using the Ahringer RNAi library clone targeting UBA-1 [54]. RNAi clones targeting CKU-80 and MLH-1 were obtained from Marc Vidal's lab [55]. HT115 bacteria carrying the empty pL4440 vector were used as control RNAi. cDNA was produced from single-worm RNA extracts using the ABscript II First synthesis system (ABclonal RK20400). RNAi effectiveness was examined by assaying the expression of the transcript being depleted in four individual animals subjected to RNAi by feeding. Expression of either the *tba-1*(F26E4.8) or *myo-3* (K12F2.1) transcripts was used as a control.

### Quantitative real time PCR (qPCR)

cDNA was produced from 3–5 worm RNA extracts using the ABscript II First synthesis (ABclonal RK20400). Real-time quantitative PCR amplifications for were carried out using ABclonal 2X SYBR Green Fast mix (Abclonal RK21200). Amplification was carried out in a LineGene 4800 (FQD48A BIOER) with initial polymerase activation at 95˚C for 2 min, followed by 40 cycles of: 95˚C for 15 sec denaturation, 60˚C for 20 sec for annealing and elongation. After 40 cycles, a melting curve analysis was carried out (60˚C to 95˚C) to verify the specificity of amplicons. Tubulin encoding *tba-1* gene was selected for a reference gene based on *C. elegans* microarray expression data [56]. Primer sequences are listed in S3 Table.

### Immunofluorescence staining

Whole-mount preparations of dissected gonads, fixation and immunostaining procedures were carried out as described in [32]. Primary antibodies were used at the following dilutions: chicken α-GFP (1:300, Abcam AB13970), mouse α-H3K4me2 (1:300, Millipore CMA-303) and rabbit α-pCHK-1 (1:200, Santa Cruz SC17922). Secondary antibodies used were: FITC anti-chicken (1:300), Cy3 anti-mouse (1:300) and Cy3 anti-Rabbit (1:250) from Jackson Immunochemicals. Immunofluorescence images were collected at 0.2 μm intervals with an Eclipse Ti2-E inverted microscope and a DSQi2 camera (Nikon). Photos were taken with a 60 x objective in combination with 1.5x auxiliary magnification and were subjected to deconvolution by using the NIS Elements software (Nikon). Partial projections of half nuclei were shown.

To examine AMX-1 histone demethylase activity *in vivo* by staining for dimethylated H3K4 level, dissected gonads from wild type, *spr-5* and *amx-1* age-matched adult hermaphrodites were probed for H3K4me2. To ensure that both control and mutant animals were processed and imaged under identical conditions, we dissected the control and mutant animals on the same slide. For this purpose, as controls we used either a wild type strain expressing GFP::

H2B, which can be easily distinguished from the mutants due to the GFP signal present throughout the dissected gonads, or wild type (N2) animals, which were fixed on the same slide but not mixed with the mutant (care was taken to keep the wild type (N2) and mutant dissected gonads on separate sections of the same slides). Fluorescent intensity signal is sampled from multiple nuclei from different gonads and quantitated by using ImageJ (Fig 4).

### Quantitative analysis of germ cell apoptosis

Germlines of age-matched (20 hours post-L4) animals were analyzed by acridine orange staining, as described in [57], utilizing a Nikon Ti2-E fluorescence microscope. Between 22 and 95 gonads were scored for each genotype. Statistical comparisons between genotypes were performed using the two-tailed Mann-Whitney test, 95% C.I.

### Supporting information

**S1 Fig. Representative amino acid sequence alignment of LSD2/AMX-1 sequences.** AMX-1 (*C. elegans* CELE_B0019.1), KDM1B (humans), SW:Q8CIG3 (mouse) and Su(*var*)3-3-PB (fly) were aligned using the Uniplot Clustal O multiple sequence alignment tool. SWIRM and histone demethylase domains are indicated. (*) denotes a single, fully conserved residue; (:) indicates conservation between groups of highly similar properties and (.) represents conservation between groups of weakly similar properties.
(TIF)

**S2 Fig. Homozygous *amx-1* mutants exhibit a reduction in brood size compared to wild type; however, brood size was not significantly reduced in the heterozygote, suggesting that *ok659* is a recessive allele.** Brood size is scored among the progeny of worms of the indicated genotypes. Error bars represent the standard error of the mean. N = 24 for each genotype. An asterisk indicates a statistically significant reduction compared to wild type by the two-tailed Mann-Whitney test, 95% C.I.
(TIF)

**S3 Fig. Brood size is not affected in *amx-1*::*GFP* transgenic animals suggesting that it expresses functional AMX-1.** The brood size of *amx-1* mutants is significantly decreased compared to wild type (P = 0.0087) while it is not altered for *amx-1*::GFP worms (P = 0.7922).
(TIF)

**S4 Fig. *amx-1*::GFP signal is not observed in embryos following *amx-1* RNAi depletion.** Asterisks indicate gut autofluorescence. Bar = 10μm.
(TIF)

**S5 Fig. AMX-1 is rarely localized in the germline.** We observed 5% of gonads containing AMX-1::GFP signal either in the premeiotic tip or pachytene stage (5 out of 100 and 5 out of 96, respectively). Wild type (N2) is used as a control. P = 0.032 in control versus AMX-1::GFP in the premeiotic tip, and P = 0.0248 in control versus AMX-1::GFP in pachytene, by the two-tailed t-test. Bars = 10μm.
(TIF)

**S6 Fig. *amx-1* mutant exhibits resistance against interstrand crosslinking agents (150 μM HN2) while no distinct resistance is observed upon lack of SPR-5 expression.** N = 70–80 worms.
(TIF)

**S7 Fig. Relative hatching observed for eggs laid by either control worms (empty vector; pL4440) or worms depleted of *mlh-1* by RNAi after treatment with the indicated doses of nitrogen mustard and cisplatin.** 8% vs 24% at 150 μM HN2, P = 0.0003, and 82% vs 95% at 300 μg/ml cisplatin, P<0.0001, by the two-tailed Mann-Whitney test, 95% C.I.
(TIF)

**S8 Fig. Co-immunostaining for RAD-51 and AMX-1::GFP in early embryos treated with HN2.** No distinct RAD-51 foci were observed in embryos in wild type control. *amx-1*::GFP animals were treated with 150 μM of HN2 in NGM containing *E. coli* OP50 for 20 hours. Higher magnification image is shown as inset. Bar = 2 μm.
(TIF)

**S9 Fig. Quantitation of RAD-51 foci in germlines of the indicated genotypes.** Mean numbers of RAD-51 foci observed per nucleus on each zone along the germline axis (x-axis). P = 0.8888 in *amx-1* and *amx-1* + *cku-80* (RNAi) in early pachytene. P = 0.1763 in *amx-1* and *amx-1* + *cku-80* (RNAi) in mid pachytene. P = 0.5509 in *amx-1* and *amx-1* + *cku-80* (RNAi) in late pachytene. P = 0.2431 in *amx-1* and *amx-1* + *cku-80* (RNAi) in diplotene by the two-tailed Mann–Whitney test, 95% C.I. Error bars represent standard error of the mean.
(TIF)

**S10 Fig. Wider field high-magnification images of whole-mounted gonads immunostained for H3K4me2.** Premeiotic tip and pachytene stage nuclei are shown for the indicated genotypes. Bars, 2 μm.
(TIF)

**S1 Table. Number of gonads exhibiting AMX-1::GFP signal observed in this study for the indicated genotypes.**
(DOCX)

**S2 Table. Number of gonads containing pCHK1 signal observed in this study for the indicated genotypes.**
(DOCX)

**S3 Table. Primer information for Gene Expression Assessed by qPCR analysis.**
(DOCX)

## Acknowledgments

We thank the Caenorhabditis Genetics Center for worm strains. We thank the members of the Kim lab for proofreading the manuscript.

## Author Contributions

**Conceptualization:** Monica P. Colaiácovo, Hyun-Min Kim.

**Data curation:** Xiaojuan Zhang, Monica P. Colaiácovo, Hyun-Min Kim.

**Formal analysis:** Sisi Tian, Sara E. Beese-Sims, Monica P. Colaiácovo, Hyun-Min Kim.

**Funding acquisition:** Monica P. Colaiácovo, Hyun-Min Kim.

**Investigation:** Xiaojuan Zhang, Jingjie Chen, Hyun-Min Kim.

**Methodology:** Monica P. Colaiácovo, Hyun-Min Kim.

**Resources:** Nara Shin, Monica P. Colaiácovo, Hyun-Min Kim.

**Supervision:** Monica P. Colaiácovo, Hyun-Min Kim.

**Validation:** Xiaojuan Zhang, Sisi Tian, Hyun-Min Kim.

**Visualization:** Xiaojuan Zhang.

**Writing – original draft:** Hyun-Min Kim.

**Writing – review & editing:** Nara Shin, Monica P. Colaiácovo, Hyun-Min Kim.

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
