## [Decision Letter · Decision Letter 0]

12 Apr 2021

Dear Dr Kim,

Thank you very much for submitting your Research Article entitled 'Histone demethylase AMX-1 provides sensitivity to interstrand crosslink DNA damage' to PLOS Genetics.

The manuscript was fully evaluated at the editorial level and by three independent peer reviewers. The reviewers appreciated the attention to an important problem, but raised some concerns about the current manuscript. Based on the reviews, we will not be able to accept this version of the manuscript, but we would be willing to review a revised version. We cannot, of course, promise publication at that time.

As you will see (below), the reviews are thoughtful and clear, so I won't paraphrase.  All of the reviewers were enthusiastic, but addressing the critiques will require (i) carefully integrating the prior literature; (ii) re-writing text and re-working some figures; and (iii) some additional experimentation.

If you decide to revise the manuscript for further consideration at PLOS Genetics, please aim to resubmit within the next 60 days, unless it will take extra time to address the concerns of the reviewers, in which case we would appreciate an expected resubmission date by email to plosgenetics@plos.org.

[LINK]

We are sorry that we cannot be more positive about your manuscript at this stage. Please do not hesitate to contact us if you have any concerns or questions.

Yours sincerely,

Gregory P. Copenhaver

Editor-in-Chief

PLOS Genetics

Gregory Barsh

Editor-in-Chief

PLOS Genetics

Reviewer's Responses to Questions

**Comments to the Authors:**

Reviewer #1: The manuscript by Zhang et al., describes the characterization of the demethylase AMX-1 in C. elegans. The findings have the potential to provide insight into the role of chromatin in DNA damage repair and ultimately how chromatin changes contribute to dysregulation in disease states. The authors find that this demethylase is primarily expressed in mitotic cells but plays a role in the germ line and promotes sensitivity to ICLs. Interestingly, the protein appears to be regulated by ubiquitination and in turn regulates the expression of the mismatch repair MutL. Overall, there is some interesting findings in this manuscript but significant rewriting and incorporation of new experiments would be important to make a compelling case for the role of this demethylase in DNA damage repair.

1) The authors describe AMX-1 as “uncharacterized” and this as the first report of this demethylase being involved in ICL repair. While it is true that this is the first report in ICL repair (I think), the study by Wang et al., 2020 (Nat Struct Mol Biol) is related and the authors need to consider these published findings in this paper. Wang et al. show that the amx-1 mutant is resistant to UV (L1 based assay) and the evidence suggests that H3K4me2 levels are important for recovery from DNA damage due to global gene expression changes. Incorporating this paper will require a number of changes throughout the manuscript.

2) I recommend changing the title – AMX-1 doesn’t “provide” sensitivity. Additionally, in the author summary: “how defective histone demethylase contributes to DNA damage repair” – please rephrase with respect to the wild-type function.

3) Figure 2: Please change the label of the Y axis of left most graph of panel A. It is currently “Larval arrest (%adult)” this is confusing. The other graphs show the defect – on this one it is the wild-type phenotype, although the label indicates the mutant phenotype.

4) Figures 3 and 4: Could the GFP signal in the gut and sheath cells be enhanced? It is very difficult to see. I am also struggling with the lack of expression in the germ line, yet the effect on H3K4me2 is evident in germ cells. It would be really nice to examine tissue-specific inactivation of AMX-1 to determine whether it functions cell autonomously/non-autonomously.

5) Figure 5: The CHK-1 signal is not very convincing – this is particularly true for the "premeiotic (misspelled) tip-TZ" as many of the foci in the amx-1 mutant do not overlap with the DAPI signal; new images and quantification need to be included. Please also quantify the “chromosome fragments” observed in the presence of crosslinking agents. I find it confusing why you see elevation of chromosome fragments but a resistance to ICLs in the mutant. This needs to be discussed. In terms of the lack of effect of UV – this will need to be addressed here given the previous publication. The authors also show that there is a reduction in mlh-1 expression in amx-1. I recommend that the authors also look at the expression of NER genes, as NER plays an important role in ICL repair in C. elegans.

6) The authors argue that AMX-1 expression compensates for lack of another demethylase, SPR-5. I think it will be important to examine the phenotypes of the amx-1; spr-5 double mutant compared to each of the single mutants to say more about the role of these demethylases in repair and whether or not they compensate for each other. As part of this, is there a progressive sterility phenotype of amx-1 as there is with spr-5?

7) The way Figure 7 is set up, suggests that you are examining RAD-51 following treatment of ICLs, but you are not. Please rewrite the set up for this experiment. The authors may want to consider examining DAPI-staining bodies in the cep-1; amx-1 double mutant to determine whether there are aberrant nuclei that are culled by apoptosis.

Reviewer #2: In this study, the authors claim that lysine-specific histone demethylase 2 (LSD2) homolog, AMX-1 provides sensitivity to DNA inter-strand crosslinks. They showed resistance against nitrogen mustard (HN2) which cause DNA inter-strand crosslinks in amx-1 mutants compared with wild-type. The mRNA expression of mismatch repair protein MLH-1 decreased in amx-1 mutants. This is reminiscent to the loss of MLH1 expression induces the resistance to DNA intra-strand crosslink caused by cisplatin in the human cancer cell line. Histone H3 K4me2 accumulates in amx-1 mutants suggesting that AMX-1 is required for demethylation of H3K4me2. While LSD2 harbors RING-like zinc finger motif and acts as an E3 ubiquitin ligase, AMX-1 does not have the zinc finger motif. But still they showed ubiquitination-dependent relocalization of AMX-1::GFP in embryos after treatment of HN2. RAD-51 foci accumulate in amx-1 mutants however they could not reach the conclusion that those are accumulations of repair intermediates or hyper activation of DNA repair machinery.

Minor Comments

p.19-20,

In the strain and alleles section, his6 follows after amx-1::gfp.

In the transgenic worms section, the authors described that GFP-HA is inserted before the stop codon of amx-1 gene. Which is correct his6 and HA?

p.17 line6

the -> the

Major Comments

p.31 Fig3, p.36 Fig6

The number of cells in the embryos is not matched in WT and spr-5 mutants at Fig3A. Similarly, the left embryo at the right panels in the amx-1::GFP+HN2 in Fig 6A looks comma stage. The stages of embryogenesis should be matched between controls and samples though the paper.

p.34 Fig5B

I suggest that the authors try to add additional doses in both camptothecin and Hydroxyurea because there are no decreases of hatching of wild-type in 500nM of CPT and 15mM HU even if some degree of decrease were observed in positive control mutants (him-18 and ztf-8 respectively).

p.10

I agree that some degrees of NH2 resistance were observed in amx-1 mutants. If you claim this was caused by DNA inter-strand crosslinks, it worth to check to use other reagents causing DNA inter-strand crosslinks, such as mitomycin C. And if the authors want to compare the phenomenon in the cancer cell line in which the expression of MLH1 decreased and C. elegans amx-1 mutants, intra-strand crosslinks inducing reagents, such as cisplatin should be used in the series of DNA damage sensitivity assay.

P.13

If the authors argue that there is HN2 resistance in amx-1 mutants, the levels of RAD-51 foci provably decrease in amx-1 mutants compared with wild-type after treatment of HN2 if RAD-51 foci indicate the accumulation of DNA damages. This particular experiment will be required to prove the resistance. Observe RAD-51 foci and AMX-1::GFP localization in the embryo after treatment of HN2.

p.17

It is also worth to check whether HN2 (and cisplatin) resistance is observed in mlh-1 mutants.

Reviewer #3: review is uploaded as an attachment

**Have all data underlying the figures and results presented in the manuscript been provided?**

Reviewer #1: Yes

Reviewer #2: Yes

Reviewer #3: Yes

PLOS authors have the option to publish the peer review history of their article (what does this mean?). If published, this will include your full peer review and any attached files.

Reviewer #1: No

Reviewer #2: No

Reviewer #3: No

---

## [Decision Letter · Decision Letter 1]

28 Jun 2021

Dear Dr Kim,

Thank you very much for submitting your Research Article entitled 'Histone demethylase AMX-1 is necessary for proper sensitivity to interstrand crosslink DNA damage' to PLOS Genetics. Reviewers 2 & 3 are now satisfied, but Reviewer 1 has identified a handful of minor concerns that will need to be addressed.  You should be able to do this easily by amending the text (but please be sure to do so fully) - once you make these changes we should not need to seek additional external peer-review before making our final editorial decision.

We therefore ask you to modify the manuscript according to the review recommendations. Your revisions should address the specific points made by each reviewer.

[LINK]

Yours sincerely,

Gregory P. Copenhaver

Editor-in-Chief

PLOS Genetics

Gregory Barsh

Editor-in-Chief

PLOS Genetics

Reviewer's Responses to Questions

**Comments to the Authors:**

Reviewer #1: The revised manuscript by Zhang et al., describes the characterization of the demethylase AMX-1 in C. elegans. The authors have done a good job revising the manuscript; however, there are still a few points that should be addressed:

1) While the authors now cite the report by Wang et al., 2020 (Nat Struct Mol Biol) there are still several places in the manuscript where they describe axm-1 as “previously uncharacterized” and “has not been investigated”. This is particularly evident in the Abstract, Author Summary, Introduction and Results. I ask that the authors carefully go through the manuscript and change these statements as appropriate.

2) In several places, the authors state that AMX-1 localizes to the embryo, gut and sheath cells. While I understand what they mean, “embryo” is not parallel to “gut and sheath cells”, please reword.

3) While the authors clearly show that inactivation of either amx-1 or spr-5 lead to elevated expression levels of the other demethylase, at several places in the manuscript they say that they compensate for each other. Expression does not equate to function and unfortunately, the authors did not examine the phenotype of the amx-1; spr-5 double mutant (except looking at H3K4me levels). In the absence of that data, I do not think the authors should say that they compensate for each other, especially since they do have several different phenotypes as documented in this paper.

Reviewer #2: The authors responded my all comments properly. I think this manuscript will be acceptable for publication.

Reviewer #3: Thank you for doing a terrific job of addressing all reviewer comments; it was a pleasure to read this manuscript!

**Have all data underlying the figures and results presented in the manuscript been provided?**

Reviewer #1: Yes

Reviewer #2: None

Reviewer #3: Yes

PLOS authors have the option to publish the peer review history of their article (what does this mean?). If published, this will include your full peer review and any attached files.

Reviewer #1: No

Reviewer #2: No

Reviewer #3: No

---

## [Editor Report · Decision Letter 2]

14 Jul 2021

Dear Dr Kim,

We are pleased to inform you that your manuscript entitled "Histone demethylase AMX-1 is necessary for proper sensitivity to interstrand crosslink DNA damage" has been editorially accepted for publication in PLOS Genetics. Congratulations!

Yours sincerely,

Gregory P. Copenhaver

Editor-in-Chief

PLOS Genetics

Gregory Barsh

Editor-in-Chief

PLOS Genetics

Comments from the reviewers (if applicable):

**Data Deposition**

http://datadryad.org/submit?journalID=pgenetics&manu=PGENETICS-D-21-00389R2

**Press Queries**

---

## [Editor Report · Acceptance letter]

28 Jul 2021

PGENETICS-D-21-00389R2 

Histone demethylase AMX-1 is necessary for proper sensitivity to interstrand crosslink DNA damage 

Dear Dr Kim, 

We are pleased to inform you that your manuscript entitled "Histone demethylase AMX-1 is necessary for proper sensitivity to interstrand crosslink DNA damage" has been formally accepted for publication in PLOS Genetics! Your manuscript is now with our production department and you will be notified of the publication date in due course.

With kind regards,

Katalin Szabo

PLOS Genetics

On behalf of:
